# Localized interlayer excitons in MoSe$_2$–WSe$_2$ heterostructures without a moiré potential

Fateme Mahdikhanysarvejahany [1], Daniel N. Shanks [1], Matthew Klein[1], Qian Wang[2], Michael R. Koehler[3], David G. Mandrus[4,5,6], Takashi Taniguchi [7], Kenji Watanabe [8], Oliver L. A. Monti[1,9], Brian J. LeRoy[1] & John R. Schaibley [1] ✉

Interlayer excitons (IXs) in MoSe$_2$–WSe$_2$ heterobilayers have generated interest as highly tunable light emitters in transition metal dichalcogenide (TMD) heterostructures. Previous reports of spectrally narrow (<1 meV) photoluminescence (PL) emission lines at low temperature have been attributed to IXs localized by the moiré potential between the TMD layers. We show that spectrally narrow IX PL lines are present even when the moiré potential is suppressed by inserting a bilayer hexagonal boron nitride (hBN) spacer between the TMD layers. We compare the doping, electric field, magnetic field, and temperature dependence of IXs in a directly contacted MoSe$_2$–WSe$_2$ region to those in a region separated by bilayer hBN. The doping, electric field, and temperature dependence of the narrow IX lines are similar for both regions, but their excitonic g-factors have opposite signs, indicating that the origin of narrow IX PL is not the moiré potential.

Localized excitons (Coulomb-bound electron–hole pairs) which can serve as single photon emitters have been investigated for decades due to their potential applications in quantum information science and optoelectronics[1–9]. Recently there has been significant interest in moiré effects in 2D material heterostructures that arise from the in-plane superlattice potential that occurs between two twisted or lattice mismatched layers[7,8,10]. IXs are spatially indirect excitons comprised of an electron in the MoSe$_2$ layer bound to a hole in the WSe$_2$ layer[2,9]. Seyler et al. were the first to report spectrally narrow PL arising from IXs in MoSe$_2$–WSe$_2$ heterostructures. In contrast, Kha et al. observed spectrally wide (-20 meV) PL peaks with alternating polarization, attributed to excited states of moiré IX in the MoSe$_2$–WSe$_2$ heterostructure[7,8]. The narrow PL lines were observable only at low (<15 K) temperatures and low (<100 nW) excitation power and consisted of an inhomogeneous distribution of narrow lines with a spread on the order of 20 meV[8]. The narrow PL lines were attributed to single IXs trapped to moiré potential

sites, as evidenced by the spectrally narrow (<1 meV) PL emission and circularly polarized optical selection rules[7,11]. There have been numerous papers studying these narrow lines since the initial report, including reports of single photon emission, charged IXs, and Coulomb staircase effects[12–14]. In this work, we show that the spectrally narrow IX PL lines are still present in MoSe$_2$-hBN-WSe$_2$ heterostructures, where the moiré potential is suppressed by an hBN spacer layer (see Supplementary Figs. 1 and 2). We compare the physical behaviors of narrow IX PL emission from non-separated, directly contacted (DC) and hBN-separated sample regions and show that the localization potential resulting in the narrow IX lines is likely due to an extrinsic disorder effect, not the moiré potential.

## Results

The sample structure is comprised of a twist-angle aligned MoSe$_2$–WSe$_2$ heterostructure as depicted in Fig. 1a (see "Methods"). In

[1]Department of Physics, University of Arizona, Tucson, AZ 85721, USA. [2]Guangdong Provincial Key Laboratory of Quantum Metrology and Sensing & School of Physics and Astronomy, Sun Yat-Sen University (Zhuhai Campus), Zhuhai 519082, China. [3]IAMM Diffraction Facility, Institute for Advanced Materials and Manufacturing, University of Tennessee, Knoxville, TN 37920, USA. [4]Department of Materials Science and Engineering, University of Tennessee, Knoxville, TN 37996, USA. [5]Materials Science and Technology Division, Oak Ridge National Laboratory, Oak Ridge, TN 37831, USA. [6]Department of Physics and Astronomy, University of Tennessee, Knoxville, TN 37996, USA. [7]International Center for Materials Nanoarchitectonics, National Institute for Materials Science, 1-1 Namiki, Tsukuba 305-0044, Japan. [8]Research Center for Functional Materials, National Institute for Materials Science, 1-1 Namiki, Tsukuba 305-0044, Japan. [9]Department of Chemistry and Biochemistry, University of Arizona, Tucson, AZ 85721, USA. ✉e-mail: johnschaibley@email.arizona.edu

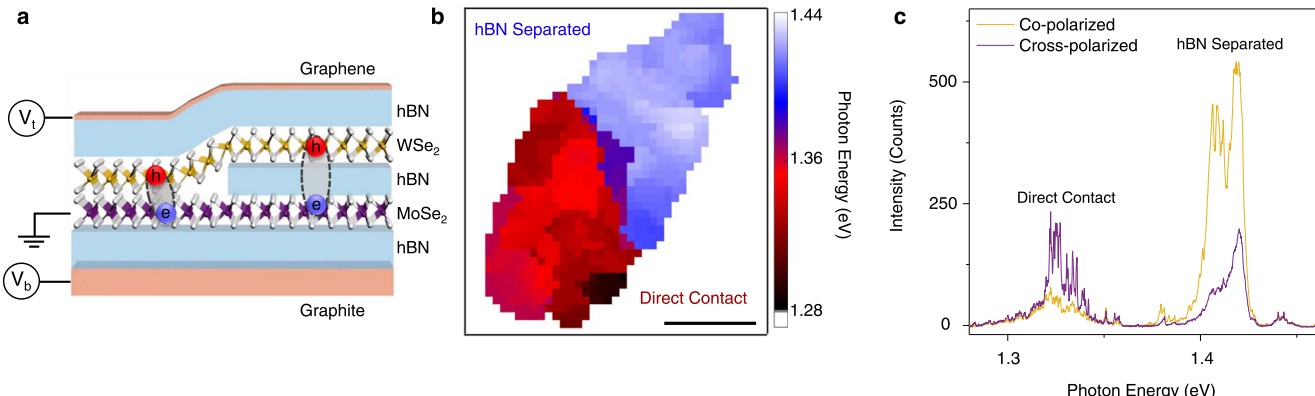

**Fig. 1 | Schematic of the sample and spatially resolved IX photoluminescence. a** Cartoon depiction of the device shows the WSe$_2$–MoSe$_2$ heterostructure encapsulated with hBN. Bilayer graphene is used for the top gate and graphite is used for the bottom gate. Approximately half of the TMD heterostructure is separated by bilayer hBN to suppress the moiré potential. **b** Confocal PL spatial map of this device, plotting the average center IX photon energy. The hBN separated region is shown in blue with 1.37 to 1.44 eV IX emission energy and the direct contact area is shown in red with the emission energy between 1.28 and 1.36 eV. **c** Co- and cross-circularly polarized PL spectra measured by exciting the hBN separated region with a polarized 1.72 eV laser and collecting from both regions at the same time. The signal from the hBN separated region ~1.42 eV is mostly co-polarized, whereas the signal from the DC region ~1.33 eV is mostly cross-polarized.

order to probe the dependence of the IX trapping on the moiré potential, we fabricated a device where half of the heterostructure has the TMD layers separated by bilayer hBN and the other half has the layers in direct contact (with no hBN spacer). The TMD layers are encapsulated in hBN, and a few layers graphene top gate ($V_t$) and graphite back gate ($V_b$) serves to independently control the charge density and external electric field experienced by the TMD layers. We used low temperature confocal PL spectroscopy to measure the spatial dependence of the IX emission photon energy (Fig. 1b). In the DC region of the device the IX photon energy is centered around 1.34 eV, consistent with R-type (near 0° twist) MoSe$_2$–WSe$_2$ heterostructures[15]. The hBN separated region shows higher energy PL centered around 1.42 eV. This 80 meV increase in the energy of the PL is in agreement with the suppression of the moiré potential by the insertion of bilayer hBN (see Supplementary Fig. 1). When the confocal pinhole was removed, PL from both regions could be detected when exciting on the higher energy hBN-separated region. Figure 1c shows co- and cross-circularly polarized PL when exciting the hBN-separated region with a 1.72 eV laser and detecting from both regions. Surprisingly, while the DC region shows the cross-circularly polarized PL, consistent with previous studies on R-type structures[5,7,15], the hBN-separated region shows mostly co-circularly polarized PL (see Supplementary Fig. 3 for similar data from another sample).

To understand the opposite selection rules in the DC and hBN-separated regions, we must consider both the effect of the moiré potential as well as the stacking dependent IX oscillator strength. The oscillator strength was calculated as a function of interlayer stacking and separation (see Supplementary Data Fig. 4). We find that that the oscillator strength is actually dominated by recombination at the $R_h^h$ site which results in primarily observing co-circularly polarized PL. However, in the DC region, the moiré potential traps IXs at the $R_h^X$ stacking site (which has the opposite selection rule) emitting cross-circularly polarized PL. Therefore, it is the trapping of IXs at $R_h^X$ sites that results in the cross-circularly polarized PL from the DC contact region, whereas the larger oscillator strength of co-polarized emission results in co-circularly polarized PL in the hBN-separated regions. Note that this does not mean that the IX is localized at the $R_h^h$ site in the hBN-separated region, rather that since this local stacking possesses the highest oscillator strength, it dominates the PL spectrum. We also emphasize that both the DC and hBN-separated regions show a distribution of spectrally narrow (< 1 meV) IX lines indicating that they do not solely originate from the moiré potential.

In order to confirm that the narrow PL emission in the hBN separated region originates from IXs, we investigated the electric field and doping dependence which was achieved by applying simultaneous top and back gate voltages. In these measurements, we again excited the hBN separated region and measured the PL from both regions. Figure 2a shows the electric field dependence of the PL while keeping the sample charge neutral[16]. In the DC region, we observe a Stark shift of 0.6 eV/(V/nm) that matches with the known dipole moment of the IX[16,17]. By adding the hBN bilayer between the TMD layers, we increase the separation between the electron and hole and consequently the IX permanent dipole moment will increase by a factor of two. Perfectly matching with our finding yielding a 1.2 eV/(V/nm) energy shift, consistent with previous reports on hBN separated IXs[18]. We also explored the IX doping dependence for both sample regions (Fig. 2b). We identify three doping regions corresponding to i-electron doped, ii-intrinsic, and iii-hole doped similar to previous reports of charged IXs[12,13,16,19]. The relatively small intrinsic doping range (−0.1 to $0.1 \times 10^{12}$ cm$^{-2}$) is consistent with a high quality device. We note that in both regions, the IX energy increases with doping, which is consistent with previous reports[13,16]. We also note that the DC region shows more prominent fine structure in its doping dependence which was previously attributed to a Coulomb staircase effect[13] (see Supplementary Fig. 5).

In order to probe the spin-valley physics of the IXs, we measured $\sigma^-$ circularly polarized PL as a function of out-of-plane magnetic field (Fig. 3a). In the DC region, we measured an exciton g-factor of $7.0 \pm 0.4$ (Fig. 3b), consistent with numerous past works on R-type MoSe$_2$–WSe$_2$ heterostructures[7,20–22]. In the hBN separated region, we measured an exciton g-factor of $-5.4 \pm 1.0$ (Fig. 3c), which has not been previously reported. We note that the opposite sign of the g-factor is consistent with our zero field circularly polarized measurements showing that the hBN separated PL has co-circularly polarized PL when pumping at 1.72 eV.

Finally, we compared the temperature-dependent behavior of both IX species to investigate the origin of localization in the heterostructure. Figure 4a, b shows the temperature-dependent PL from both regions of the heterostructure. In Fig. 4a, we see the IX emission from the DC region of the heterostructure. By increasing the temperature, the IX emission changes from a series of individual, narrow peaks to become a homogeneous broad peak around 13 K. The behavior of the hBN-separated region shows a disappearance of the narrow IX lines at almost the same temperature (9 K); however, no broad peak persists to higher temperature. In both cases, the narrow lines disappear around

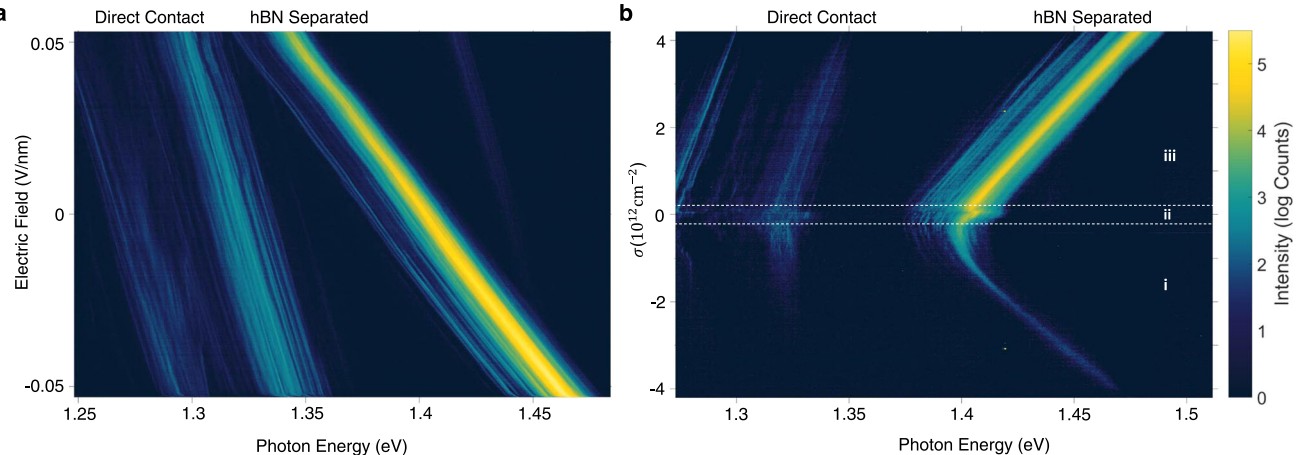

**Fig. 2 | IX photoluminescence as a function of electric field and doping level. a** PL emission as a function of electric field measured by exciting the hBN separated region while collecting from both regions. The hBN separated IX lines show a larger dipole moment due to the increased electron-hole separation. **b** PL emission as a function of doping level. The regions i, ii, and iii correspond to electron doping, neutral, and hole doping respectively.

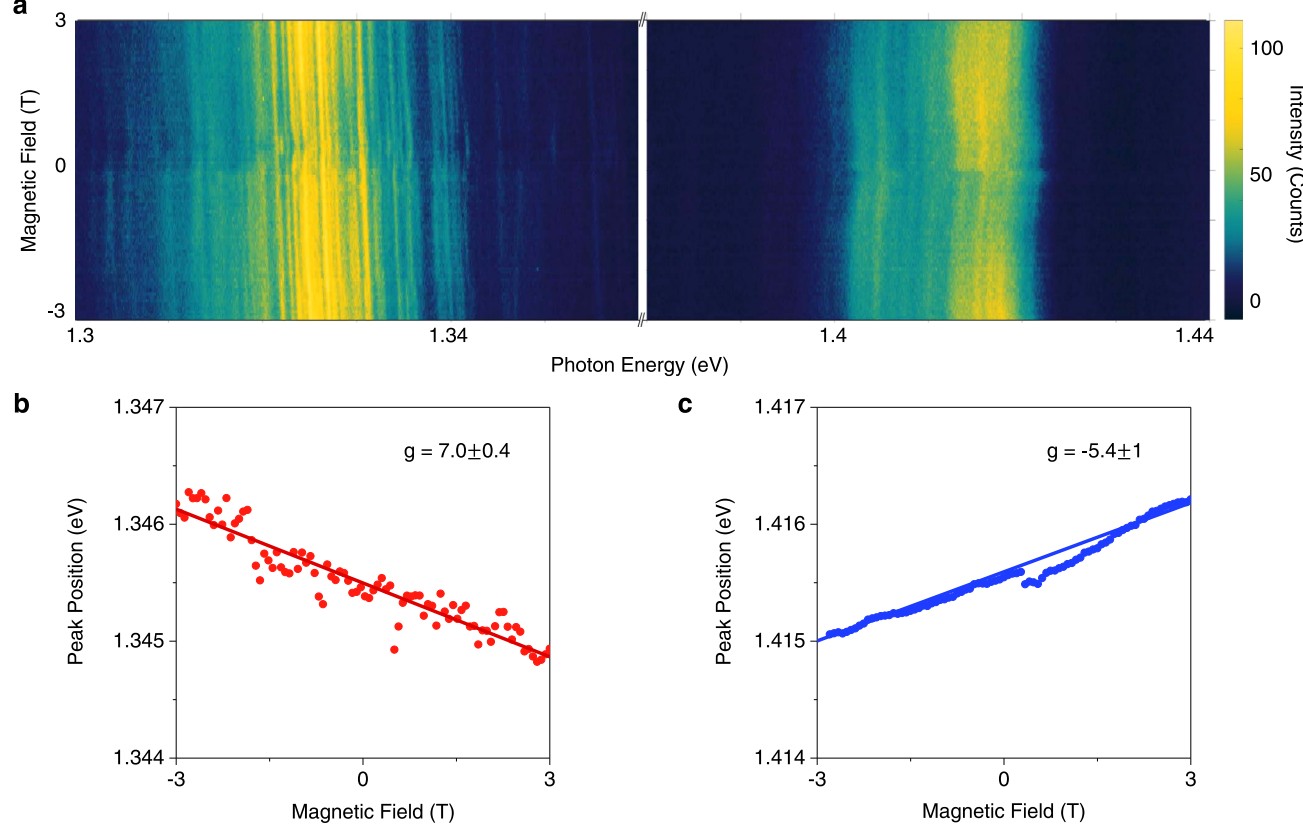

**Fig. 3 | Photoluminescence as a function of magnetic field. a** Magnetic field dependent PL (detecting $\sigma^-$) shows opposite sign of g-factors for the DC and hBN separated regions. Example magnetic field dependence of a single IX line for both DC (**b**) and hBN separated regions (**c**). The excitonic g-factors reported are average values of the fitting to six single IX lines. The error bar shows one standard deviation.

10 K, suggesting that the localization potential that gives rise to the narrow lines is same for both regions and independent of the moiré potential. We note, however, that there are differences between the two sample regions. In the DC region, a broad PL peak persists to temperatures above 19 K, whereas in the hBN separated region the PL disappears completely (Fig. 4c). See Supplementary Fig. 6 for higher temperatures. The temperature-dependent PL of both regions were measured in the same thermal cycle, using the same conditions including exposure time, power and excitation wavelength. The measurement was repeated several times with similar results. We, therefore, present a simple picture that explains all of the observed behaviors. In the DC region, a weak extrinsic localization potential sits on top of the deep -50–100 meV moiré potential (Fig. 4d). The moiré potential explains the lower IX energy in the DC region, the cross-circularly polarized PL, and the positive g-factor; even so, the narrow lines originate from the shallower extrinsic potential which disappears abruptly at -10 K. However, the broader PL signal persists to higher temperature due to the trapping of IX by the wider, deeper moiré potential. Whereas, on the hBN-separated region, the moiré potential is highly suppressed (Fig. 4d), explaining the higher IX energy, co-

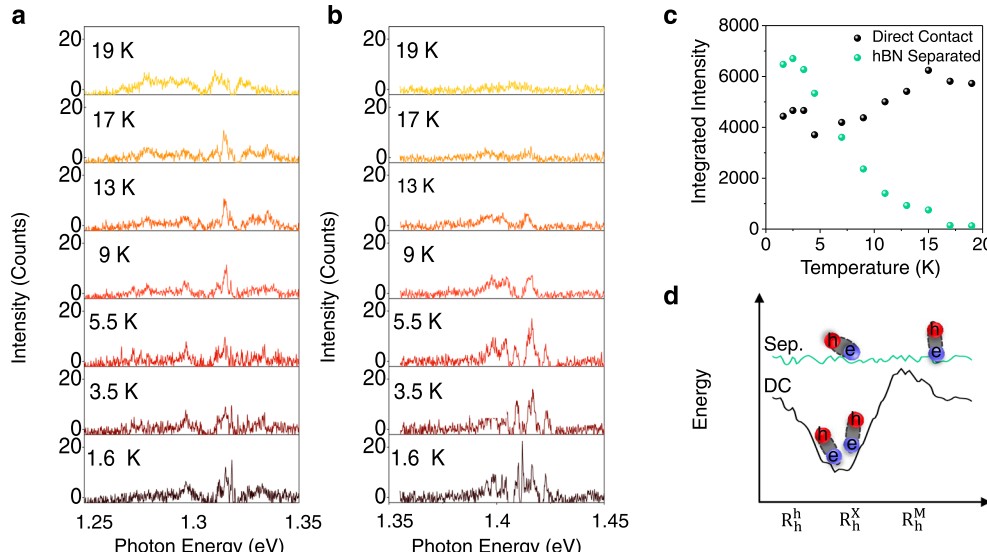

**Fig. 4 | Temperature-dependent PL and origin of narrow IX lines. a** PL of the narrow IX lines in the DC region as a function of temperature shows the width of individual lines are increasing by 9 K and disappear completely by 17 K. The signal in the DC region includes narrow emission on top of a wider PL plateau. The wider emission persists to higher temperature (see Supplementary Fig. 6) that is consistent with previous temperature-dependent measurements on R-type heterostructures. **b** Temperature-dependent PL from the hBN separated region shows the narrow lines are widening by 9 K which is in good agreement with the DC region's temperature-dependent PL. The hBN separated IX signal disappears fully by 13 K and does not have a wider PL plateau. **c** Spectrally integrated PL shows IX emission in the hBN separated region disappears completely by 19 K whereas the signal from the DC region is approximately constant. **d** Depiction of the IX energy as a function of position in the moiré, for DC region (black) and hBN separated (Sep.) region (green). Both regions exhibit a weak extrinsic trapping potential denoted by the fluctuations. The DC region has both moiré trapping and extrinsic fluctuations.

polarized optical selection rule, and negative g-factor. In this region, localization is solely due to the extrinsic potential. As such, the narrow lines are observed below 10 K, but the PL vanishes completely above 19 K since the hBN-separated region does not have the moiré potential to confine the IXs, and the strong dipole−dipole and exchange interaction scatters the IXs outside of the detection area or light cone.

## Discussion

We note that we considered the possibility that the trapping potential responsible for the narrow lines in the DC region is from the moiré potential with inhomogeneous broadening. If this were the case, the potential depth of each moiré trap should be about the same with some small variations. Theoretical predications and STM results have shown that the depth of the moiré potential in direct contact R type MoSe$_2$−WSe$_2$ heterostructures is on the order of -50−100 meV[5,23], whereas the moiré potential in the hBN separated region is considered to be negligible as evidenced by our DFT calculation (Supplementary Fig. 1). Our temperature dependent data shows that in both regions, the narrow IX PL disappears at 9 K, indicating that the depth of the potential causing the narrow IX lines is nearly the same in both regions. Based on this observation, we conclude that the nanoscale localization potential giving rise to the narrow lines in the DC region cannot be the intrinsic moiré potential. We also considered the possibility that the narrow lines originate from atomic reconstruction[24]. Previous works have shown that atomic reconstruction in R-type heterostructures exhibits two domains with $R_h^X$ and $R_h^M$ type stacking. Therefore, the selection rule of the DC region could be explained either by trapping via the moiré potential or atomic reconstruction. Future studies performing low-frequency Raman scattering may be able to distinguish these effects. However, the same arguments that apply to the moiré potential can be applied to atomic reconstruction since the hBN spacer would suppress both effects.

In summary, we have shown that the spectrally narrow IX lines that were previously attributed to intrinsic trapping via the moiré potential are extrinsic in nature and originate from nanoscale defects or nanobubbles formed during the 2D heterostructure fabrication process. In DC heterostructures this extrinsic potential sits on top of the moiré potential giving rise to narrow IX lines that exhibit the characteristics of intrinsic moiré IXs. Our result provides crucial insights into future quantum device applications of moiré excitons and motivates the development of improved 2D material fabrication techniques to realize the goal of 2D quantum emitter arrays with homogeneous energies. Our results demonstrate that spatially trapped IXs are required to obtain optically detectable PL. The disappearance of PL in the hBN separated region as the IXs become delocalized is evidence of this.

## Methods
### Sample fabrication
The layers of WSe$_2$, MoSe$_2$, hBN and graphene were exfoliated from bulk using the scotch tape method. The layer thickness was measured by atomic force microscopy and optical contrast. For aligning the TMD monolayers near 0° degree precisely, we used polarization resolved second harmonic generation spectroscopy[25,26]. The device was fabricated using the dry transfer technique[27]. The bottom and top hBN thicknesses were 22 and 8 nm respectively. The top gate was bilayer graphene, and the bottom gate was 2 nm thick graphite. 7 nm/40 nm chrome gold contacts to the device were patterned using electronic beam lithography and thermal evaporation.

### Optical measurements
For the confocal and polarized PL measurements, we used a 1.72 eV photon energy continuous wave laser (M Squared SOLSTIS) on resonance with WSe$_2$. Unless otherwise noted the excitation power was 20 μW and the sample temperature was 1.6 K. The experiments were performed in the reflection geometry focusing the laser and collecting with a 0.81 NA attocube objective. In the confocal measurements, a detection area of 1 μm was achieved by spatially filtering the PL and using a 50 μm pinhole and a 50× magnification confocal setup. We used appropriate combinations of polarizers and achromatic waveplates to control the excitation and detection polarizations.

## Electronic control

The electric field reported in Fig. 2a is calculated by $\mathbf{E_{hs}} = \left(\frac{V_t - V_b}{t_t + t_b}\right) * \frac{\varepsilon_{hBN}}{\varepsilon_{hs}}$ where $V_t$ ($V_b$) is the voltage applied to the top (bottom) gate, $t_{t(b)}$ is the thickness of the top (bottom) hBN, $\varepsilon_{hBN} = 3.7$ ($\varepsilon_{hs}$) is the relative dielectric constant of the hBN (heterostructure). The dielectric constant of the heterostructure was determined by taking the weighted average (weighted by layer thickness) of the dielectric constant of the TMD and hBN layers. This is calculated to be $\varepsilon_{hs} = \{(t_{WSe_2} * \varepsilon_{WSe_2}) + (t_{hBN} * \varepsilon_{hBN}) + (t_{MoSe_2} * \varepsilon_{MoSe_2})\}/(t_{WSe_2} + t_{hBN} + t_{MoSe_2}) = 6.26$ where $t_{WSe_2}$, $t_{MoSe_2}$ and $t_{hBN}$ are the thickness of the WSe$_2$, MoSe$_2$, and hBN layers respectively. The doping density is calculated using the parallel plate capacitor model where $\sigma = \varepsilon_{hBN}(V_b + V_t)/t_{total}$.

## DFT Calculation

We create a $1 \times 1$ MoSe$_2$/WSe$_2$ unit cell with a lattice constant of 3.317 Å, which is the average of the MoSe$_2$ and WSe$_2$ lattice constants. All calculations of the K point bandgap ($E_g$) of MoSe$_2$/WSe$_2$ hetero-bilayers with different translation $\mathbf{r}_0$ between layers are performed in the framework of density functional theory by using a plane-wave basis set as implemented in the Vienna ab initio simulation package. For each given $\mathbf{r}_0$, we fix the interlayer distance $d = 6.447$ Å, which is the minimum value among all given $\mathbf{r}_0$ in our calculations. We calculate $E_g$ for this interlayer distance along with a series of increasing interlayer distances of 6.547, 6.647, 6.747, 6.847, 6.947, 7.447, and 9.447 Å. These increasing distances simulate the effect of separating the layers with the addition of hBN.

The electron-ion interactions are modeled using the projector augmented wave (PAW) potentials. The generalized gradient approximation (GGA) with the Perdew-Burke-Ernzerhof (PBE) functional with van der Waals corrections (vdW) for the exchange-correlation interactions is used. In all of our calculations the spin-orbit coupling (SOC) is fully taken into account. A vacuum of 15 Å is used in all the calculations to avoid interaction between the neighboring slabs. The plane-wave cutoff energy is set to 500 eV and the first Brillouin zone of the unit cell of MoSe$_2$/WSe$_2$ hetero-bilayers is sampled by using the Monkhorst–Pack scheme of $k$-points with the $12 \times 12 \times 1$ mesh for the structural optimization and the $24 \times 24 \times 1$ mesh for the band structure. The residual forces have converged to less than 0.01 eV/Å and the total energy difference to less than $10^{-5}$ eV.

## Data availability

The data that support the findings of this study are available in the Figshare database at the following links: https://figshare.com/projects/Localized_Interlayer_Excitons_in_MoSe2-WSe2_Heterostructures_without_a_Moir_Potential/146136.

## Code availability

The code that support the findings of this study are available in the Figshare database at the following links: https://figshare.com/projects/Localized_Interlayer_Excitons_in_MoSe2-WSe2_Heterostructures_without_a_Moir_Potential/146136.

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

## Acknowledgements

We acknowledge Hongyi Yu for useful discussions. J.R.S. and B.J.L. acknowledge support from the National Science Foundation Grant. Nos. DMR-2003583 and ECCS-2054572. J.R.S. acknowledges support from Air Force Office of Scientific Research Grant Nos. FA9550-20-1-0217 and FA9550-21-1-0219. B.J.L. acknowledges support from the Army Research Office under Grant nos. W911NF-18-1-0420 and W911-NF-20-1-0215. D.G.M. acknowledges support from the Gordon and Betty Moore Foundation's EPiQS Initiative, Grant GBMF9069. K.W. and T.T. acknowledge support from JSPS KAKENHI Grant Nos. 19H05790, 20H00354 and 21H05233.

## Author contributions

J.R.S. and B.J.L. conceived and supervised the project. D.N.S. fabricated the structures and F.M. performed the experiments, assisted by D.N.S. and M.K. F.M. analyzed the data with input from D.N.S., J.R.S., and B.J.L. M.R.K. and D.G.M. provided and characterized the bulk $MoSe_2$ and $WSe_2$ crystals. T.T. and K.W. provided the bulk hBN crystals. F.M., D.N.S., J.R.S., and B.J.L. wrote the paper with input from O.L.A.M., T.S. and R.L.N. All authors discussed the results.

## Competing interests

The authors declare no competing interests.
