## [Peer Review File · Nature Communications]

Reviewers' Comments:

Reviewer #1:

Remarks to the Author:

Fateme et al. discuss a topic under debate since 2019 on the moiré excitons in MoSe₂/WSe₂ heterostructure and make a convincing conclusion on this topic. The debate arises due to the different findings in the back-to-back papers by Seyler et al. (Nature 567(7746): 66-70) and Kha et al. (Nature 567(7746): 71-75) on moiré excitons in MoSe₂/WSe₂ heterostructure, and in the following works there are supporters for both sides, which leaves the moiré excitons in MoSe₂/WSe₂ heterostructure controversial and mysterious. This work reaches a sound and clear conclusion, which rules out moiré potential as the origin of spectral sharp peaks discovered in the heterostructure. Therefore, I think this work is certainly of great interest to the community and of potential high impact, and I support the publication of this paper on Nature Communications after addressing several minor questions.

1. The authors never discuss or comment Kha et al.'s work in both the introduction and discussion part in this manuscript (Ref. 3 in the manuscript), which may hurt the significance of this paper. I think it will be better if there are some further comments on both sides of the debate.
2. There are several typos in the manuscript, such as "theorder" in Ln. 53 and "originof" in Ln. 119, please check the manuscript carefully.

Reviewer #2:

Remarks to the Author:

The paper aims at clarifying the physical origin of the sharp photoluminescence (PL) lines related to interlayer excitons (IXs) in moiré structures in MoSe₂-WSe₂ heterobilayers. The authors fabricated a MoSe₂-WSe₂ heterobilayers with a near 0° degree twist angle and inserted a hexagonal BN (hBN) spacer in half of the sample. They then performed cross-polarized and co-polarized PL measurements with pumping in the hBN inserted region and detection in both the hBN-inserted region and direct-contact region. Similar sharp IX PL lines are observed, and various effects including doping, electric field, and temperature on these narrow IX lines are investigated. Based on their studies they conclude that "the spectrally narrow IX lines that were previously attributed to intrinsic trapping via the moiré potential are extrinsic in nature and originate from nanoscale defects or nanobubbles formed during the 2D heterostructure fabrication process".

I think the research studies are carefully done, and their results are of high interest to the scientific community in moiré-structure related research. However, I found a few issues that need to be addressed before the paper can be considered for publication in Nature Communications. My concerns are described below.

In the introductory paragraph, the authors wrote:

"Previous reports of spectrally narrow (<1 meV) photoluminescence (PL) emission lines at low temperature have been attributed to IXs localized by the moiré potential between the TMD layers^{4,5,6,7,8,9,10}. We show that spectrally narrow IX PL lines are present even when the moiré potential is suppressed by inserting a bilayer hexagonal boron nitride (hBN) spacer between the TMD layers."

This statement implies that the assignment of sharp IX lines to localized excitons trapped in moiré potential in the above references is incorrect. However, I checked these references and only found such a statement mentioned in Ref. 10 [Nature Technology, 7, 227 (2022)] as quoted below.

"Seyler and co-workers also investigated MoSe₂/WSe₂ heterobilayers but focused on narrow resonances with a linewidth of ~80 μeV (Fig. 3b), observable at a lower temperature (~2K) and lower excitation power (~10–100 nW) compared with those presented in Fig. 3a.... Both sharp resonances and the broader interlayer exciton resonances have been observed in more recent experiments in the same MoSe₂/WSe₂ bilayers. Bai and collaborators observed both a few sharp resonances (labelled as type I in Fig. 3c) and a broad peak (type II in Fig. 3c) that evolved into multiple, broader interlayer exciton resonances with increasing excitation power"

It is well known that defects can localize an exciton and make its PL line extremely sharp (similar to an exciton trapped in an isolated quantum dot). A moiré superlattice behaves like a quantum-

dot array that has miniband structures. Therefore, the localization effect is not as strong as a defect, and the exciton PL line should not be as sharp. However, in realistic moiré structures, the periodicity is not perfect. So, there will always be localization sites associated with any given moiré pattern with incommensurate lattice matching. The exciton trapped at these localization sites can still lead to very sharp PL lines assigned as type I in Ref. 10, while the broader PL lines are associated with excitons trapped in areas with a close-to-periodic pattern, which are assigned as type II in Ref. 10. So, one cannot conclude that the sharp IX lines are solely caused by the extrinsic defects, although there will be many sharp IX lines contributed by defects when the sample is not clean. Furthermore, inserting a BN slab in half of the bilayer sample will also introduce more defects.

So, I suggest the authors revise their claim "that the localization potential resulting in the narrow IX lines is likely due to an extrinsic disorder effect, not the moiré potential" and discuss more possibilities for the assignment of sharp IX PL lines in moiré structures.

The exciton g-factor of -5.4 ± 1.0 is obtained in the hBN separated region, which is new. (Figure 3c) The authors should discuss the physical origin of the sign change of the g factor from $+7.0$ in the DC region (Figure 3b) to -5.4 in the hBN separated region. There are some studies of the variation of g factors in moiré structures of different twist angles or different stackings. An insightful discussion on that will be quite helpful.

The experimental studies were carried out with pumping in the hBN separated region and detection on both sides. So the carriers must travel through the junctions before they can recombine in the DC region. It is possible that many sharp IX PL lines are due to carriers recombining in the DC region close to the junction where the density of extrinsic defect may be high. The authors should investigate the situation with pumping in the DC region away from the junction and collect in the same region to verify if the PL spectra will remain essentially the same as their current measurements. This can provide further insight into this investigation.

Reviewer #3:

Remarks to the Author:

The authors present an experimental study on interlayer excitons (ILE) in a MoSe₂-WSe₂ heterostructure at low temperatures. Their sample structure pushes the state of the art to a new level: it contains top and bottom graphene gates insulated by hBN layers, the two TMDC layers are aligned based on second-harmonic generation spectroscopy, and a bilayer hBN flake is partially inserted to act as a spacer between the TMDC layers. This spacer should also act to suppress the moiré potential that arises from the coupling between the TMDC layers.

In their optical spectroscopy experiments, the authors observe multiple, spectrally narrow (1 meV) lines in both, the directly contacted (DC) and the hBN-separated regions of their heterostructure at temperatures below about 10 K. Previously, such emission lines had been associated with ILE trapped in the moiré potential formed in near-aligned TMDC heterostructures. Gate-dependent measurements nicely reveal a linear Stark shift of these lines, with a significantly larger electric dipole moment in the hBN-separated region as expected from the larger electron-hole separation. Circularly polarized excitation shows that in the DC region, the ILE emission is cross-polarized, while in the separated region, the ILE emission is co-polarized. Aligned with this observation, the effective g factors observed in the two regions have opposite sign, but comparable magnitude, indicating a change of selection rules.

Based on the fact that they observe spectally narrow ILE emission from both regions, the authors infer that the moiré potential cannot be the sole cause for a trapping potential. Instead, they propose that extrinsic effects, such as nanoscale defects caused by sample fabrication, lead to an additional trapping potential.

While I find the experimental results beautiful and interesting, I have a number of questions regarding their interpretation.

Given that the authors aim for 0 degree alignment, I find it peculiar that they do not discuss effects such as atomic reconstruction, which has been shown to take place in TMDC heterostructures at low twist angles. This can provide domains of well-defined interlayer stacking that could also act as a trapping potential and may lead to emission being dominated by a single

type of high-symmetry alignment. Can the authors determine (e.g., by low-frequency Raman spectroscopy) whether they have atomic reconstruction? In order to study the impact of atomic reconstruction vs. moiré on ILE trapping, measurements on samples with controlled mis-alignment on the order of 3-5 degrees would be enlightening.

It is not clear to me why the ILE emission in the separated part of the flake should be co-polarized to such a high degree, if localization is provided by a random potential: why would this favor emission from, e.g., R_{hh} high-symmetry points, which would have the right selection rules and a roughly fitting g factor based on theory calculations (Ref. 21)?

I was also surprised by the suppression of ILE PL in the separated region at slightly higher temperatures. Given the low excitation density used by the authors, I wonder if dipole-dipole interaction scattering the ILE out of the detection spot can really be the driving factor for this observation. This could be verified by performing power-dependent PL measurements on the separated region and imaging the emission spot size as a function of power.

At the current stage, I do not feel that the manuscript provides fully convincing evidence to support the authors claims, but a suitable revision addressing the above-mentioned points could be of great interest to the 2D materials community and warrant publication in Nature communications.

Response to Reviewers (NCOMMS-22-18201-T)

Reviewer #1 (Remarks to the Author):

Comment 1.1) Fateme et al. discuss a topic under debate since 2019 on the moiré excitons in MoSe₂/WSe₂ heterostructure and make a convincing conclusion on this topic. The debate arises due to the different findings in the back-to-back papers by Seyler et al. (Nature 567(7746): 66-70) and Kha et al. (Nature 567(7746): 71-75) on moiré excitons in MoSe₂/WSe₂ heterostructure, and in the following works there are supporters for both sides, which leaves the moiré excitons in MoSe₂/WSe₂ heterostructure controversial and mysterious. This work reaches a sound and clear conclusion, which rules out moiré potential as the origin of spectral sharp peaks discovered in the heterostructure. Therefore, I think this work is certainly of great interest to the community and of potential high impact, and I support the publication of this paper on Nature Communications after addressing several minor questions.

Response 1.1) We thank the reviewer for their constructive review and for stating that “this work is certainly of great interest to the community and of potential high impact.”

Comment 1.2) The authors never discuss or comment Kha et al.’s work in both the introduction and discussion part in this manuscript (Ref. 3 in the manuscript), which may hurt the significance of this paper. I think it will be better if there are some further comments on both sides of the debate.

Response 1.2) We thank the reviewer for the suggestion and have added a discussion of Kha et al.’s. work to the manuscript.

Comment 1.3) There are several typos in the manuscript, such as “theorder” in Ln. 53 and “originof” in Ln. 119, please check the manuscript carefully.

Response 1.3) We have carefully edited the revised manuscript and appreciate the reviewer for pointing out these typos.

Reviewer #2 (Remarks to the Author):

Comment 2.1) The paper aims at clarifying the physical origin of the sharp photoluminescence (PL) lines related to interlayer excitons (IXs) in moiré structures in MoSe₂-WSe₂ heterobilayers. The authors fabricated a MoSe₂-WSe₂ heterobilayers with a near 0° degree twist angle and inserted a hexagonal BN (hBN) spacer in half of the sample. They then performed cross-polarized and co-polarized PL measurements with pumping in the hBN inserted region and detection in both the hBN-inserted region and direct-contact region. Similar sharp IX PL lines are observed, and various effects including doping, electric field, and temperature on these narrow IX lines are investigated. Based on their studies they conclude that “the spectrally narrow IX lines that were previously attributed to intrinsic trapping via the moiré potential are extrinsic in nature and originate from nanoscale defects or nanobubbles formed during the 2D heterostructure fabrication process”.

I think the research studies are carefully done, and their results are of high interest to the scientific community in moiré-structure related research. However, I found a few issues that need to be addressed before the paper can be considered for publication in Nature Communications. My concerns are described below.

Response 2.1) We thank the reviewer for preparing a thoughtful review which has improved our manuscript.

Comment 2.2) In the introductory paragraph, the authors wrote: “Previous reports of spectrally narrow (<1 meV) photoluminescence (PL) emission lines at low temperature have been attributed to IXs localized by the moiré potential between the TMD layers^{4,5,6,7,8,9,10}. We show that spectrally narrow IX PL lines are present even when the moiré potential is suppressed by inserting a bilayer hexagonal boron nitride (hBN) spacer between the TMD layers.” This statement implies that the assignment of sharp IX lines to localized excitons trapped in moiré potential in the above references is incorrect. However, I checked these references and only found such a statement mentioned in Ref. 10 [Nature Technology, 7, 227 (2022)] as quoted below. “Seyler and co-workers also investigated MoSe₂/WSe₂ heterobilayers but focused on narrow

resonances with a linewidth of $\sim 80\mu\text{eV}$ (Fig. 3b), observable at a lower temperature ($\sim 2\text{K}$) and lower excitation power ($\sim 10\text{--}100\text{nW}$) compared with those presented in Fig. 3a.... Both sharp resonances and the broader interlayer exciton resonances have been observed in more recent experiments in the same MoSe₂/WSe₂ bilayers. Bai and collaborators observed both a few sharp resonances (labelled as type I in Fig. 3c) and a broad peak (type II in Fig. 3c) that evolved into multiple, broader interlayer exciton resonances with increasing excitation power”

Response 2.2) We thank the reviewer for catching this mistake. We have corrected the references to match with the content of the papers.

Comment 2.3) It is well known that defects can localize an exciton and make its PL line extremely sharp (similar to an exciton trapped in an isolated quantum dot). A moiré superlattice behaves like a quantum-dot array that has miniband structures. Therefore, the localization effect is not as strong as a defect, and the exciton PL line should not be as sharp. However, in realistic moiré structures, the periodicity is not perfect. So, there will always be localization sites associated with any given moiré pattern with incommensurate lattice matching. The exciton trapped at these localization sites can still lead to very sharp PL lines assigned as type I in Ref. 10, while the broader PL lines are associated with excitons trapped in areas with a close-to-periodic pattern, which are assigned as type II in Ref. 10. So, one cannot conclude that the sharp IX lines are solely caused by the extrinsic defects, although there will be many sharp IX lines contributed by defects when the sample is not clean. Furthermore, inserting a BN slab in half of the bilayer sample will also introduce more defects.

So, I suggest the authors revise their claim “that the localization potential resulting in the narrow IX lines is likely due to an extrinsic disorder effect, not the moiré potential” and discuss more possibilities for the assignment of sharp IX PL lines in moiré structures.

Response 2.3) We agree that there are several possible effects that could result in the trapping of excitons such as the moiré potential, defects, strain, and nano bubbles. Our temperature dependence shows that the narrow lines in both direct contact and hBN separated regions disappear

at the same temperature, indicating that the depth of the trapping potential in both regions is the same.

Let's consider the possibility that the trapping potential responsible for the narrow lines in the direct contact region is from the moiré potential with inhomogeneous broadening. If this were the case, the potential depth of each moiré trap should be about the same with some small variation. Theoretical predictions and STM results have shown that the depth of the moiré potential in direct contact R type MoSe₂-WSe₂ heterostructures is on order of ~50-100 meV, whereas the moiré potential in the hBN separated region is considered to be negligible as evidenced by our DFT calculation (Extended Data Fig. 1). Our temperature dependent data show that in both regions, that the narrow IX PL disappears at 9 K, indicating that the depth of the potential causing the narrow IX lines is nearly the same in both regions. Based on this observation, we conclude that the nanoscale localization potential giving rise to the narrow lines in the direct contact region cannot be the intrinsic moiré potential. We have revised the discussion of this point in the main text and thank the reviewer for pointing out this possible source of confusion.

Comment 2.4. The exciton g-factor of -5.4 ± 1.0 is obtained in the hBN separated region, which is new. (Figure 3c) The authors should discuss the physical origin of the sign change of the g factor from +7.0 in the DC region (Figure 3b) to -5.4 in the hBN separated region. There are some studies of the variation of g factors in moiré structures of different twist angles or different stackings. An insightful discussion on that will be quite helpful.

Response 2.4) To address this question, we calculated the oscillator strength as a function of interlayer stacking and separation (see Extended Data Fig.4). To understand the opposite selection rule and the sign of the g-factor in the DC and hBN-separated regions, we must consider both the effect of the moiré potential as well as the stacking dependent IX oscillator strength. In general, we find that that the oscillator strength is actually dominated by recombination at the R_h^h which results in primarily observing co-circularly polarized PL. However, in the DC region, the moiré potential traps IXs at the R_h^X stacking site which has the opposite selection rule, emitting cross-circularly polarized PL. Therefore, it is the trapping of IX at R_h^X sites that results in the cross-circularly polarized PL from the DC contact region, whereas the larger oscillator strength of co-polarized emission results in co-circularly polarized PL in the hBN-separated regions. Note that

this does not mean that the IX is localized at the R_h^h site in the hBN-separated region, rather that since this local stacking possesses the highest oscillator strength, it will dominate the PL spectrum.

The negative sign of the g factor in the hBN separated region is consistent with the co-circularly polarized optical selection rule of the IX obtained at zero magnetic field. The sign of the g factor (and related polarization selection rule) is determined by the local stacking symmetry of the two TMD layers regardless of the strength of the moiré potential. As for the magnitude of the g-factor, there have been numerous experimental and theoretical works showing that the magnitude of the g factor in direct contacted (R-type) samples is on order of +5 to +7^{1,2,3,4}.

Comment 2.5) The experimental studies were carried out with pumping in the hBN separated region and detection on both sides. So the carriers must travel through the junctions before they can recombine in the DC region. It is possible that many sharp IX PL lines are due to carriers recombining in the DC region close to the junction where the density of extrinsic defect may be high. The authors should investigate the situation with pumping in the DC region away from the junction and collect in the same region to verify if the PL spectra will remain essentially the same as their current measurements. This can provide further insight into this investigation.

Response 2.5) At the reviewer's suggestion, we performed measurements to confirm the spatial origin of these lines. We used a confocal pinhole to isolate excitation and PL emission from the middle of each region (2 μm effective diameter). Figure R1 shows PL spectra from each region.

Figure R1: IX photoluminescence in separate regions using confocal pinhole. **a**, PL emission of the hBN separated region using 1.74 eV laser with 10 nW power and the confocal pinole (2 μm resolution) shows an inhomogeneous distribution of narrow lines consistent with the measurement reported without pinhole. **b**, PL spectrum taken from the middle of the direct contact region (red) and hBN separated area (blue) using confocal spectroscopy with higher power showing the high energy signal originates in the hBN separated region while the low energy signal originates from the DC area .The data is taken with 50 μW laser power. **c**, PL spectrum of the DC area with the pinhole, we used 100 nW 1.74 eV laser.

Reviewer #3 (Remarks to the Author):

Comment 3.1) The authors present an experimental study on interlayer excitons (ILE) in a MoSe₂-WSe₂ heterostructure at low temperatures. Their sample structure pushes the state of the art to a new level: it contains top and bottom graphene gates insulated by hBN layers, the two TMDC layers are aligned based on second-harmonic generation spectroscopy, and a bilayer hBN flake is partially inserted to act as a spacer between the TMDC layers. This spacer should also act to suppress the moiré potential that arises from the coupling between the TMDC layers.

In their optical spectroscopy experiments, the authors observe multiple, spectrally narrow (1 meV) lines in both, the directly contacted (DC) and the hBN-separated regions of their heterostructure at temperatures below about 10 K. Previously, such emission lines had been associated with ILE

trapped in the moiré potential formed in near-aligned TMDC heterostructures. Gate-dependent measurements nicely reveal a linear Stark shift of these lines, with a significantly larger electric dipole moment in the hBN-separated region as expected from the larger electron-hole separation. Circularly polarized excitation shows that in the DC region, the ILE emission is cross-polarized, while in the separated region, the ILE emission is co-polarized. Aligned with this observation, the effective g factors observed in the two regions have opposite sign, but comparable magnitude, indicating a change of selection rules.

Based on the fact that they observe spectally narrow ILE emission from both regions, the authors infer that the moiré potential cannot be the sole cause for a trapping potential. Instead, they propose that extrinsic effects, such as nanoscale defects caused by sample fabrication, lead to an additional trapping potential.

While I find the experimental results beautiful and interesting, I have a number of questions regarding their interpretation.

Response 3.1) We thank the reviewer for their encouragement stating that our results are “beautiful and interesting” as well as “push[ing] the state of the art to a new level.”

Comment 3.2. Given that the authors aim for 0 degree alignment, I find it peculiar that they do not discuss effects such as atomic reconstruction, which has been shown to take place in TMDC heterostructures at low twist angles. This can provide domains of well-defined interlayer stacking that could also act as a trapping potential and may lead to emission being dominated by a single type of high-symmetry alignment. Can the authors determine (e.g., by low-frequency Raman spectroscopy) whether they have atomic reconstruction? In order to study the impact of atomic reconstruction vs. moiré on ILE trapping, measurements on samples with controlled mis-alignment on the order of 3-5 degrees would be enlightening.

Response 3.2) We thank the reviewer for these suggestions. Previous works have shown that atomic reconstruction in R-type heterostructures exhibits two domains with R_h^X and R_h^M type stacking. Therefore, the selection rule of the DC region could be explained either by trapping via

the moiré potential or atomic reconstruction⁵. We agree that the proposed study of the twist angle dependent Raman scattering would be an interesting future work and have made a reference to a recent work studying this effect to the main text. Unfortunately, at present, we do not have the ability to perform low-frequency Raman spectroscopy and cannot distinguish these effects clearly. We have added a discussion of these points to the main text.

Comment 3.3) It is not clear to me why the ILE emission in the separated part of the flake should be co-polarized to such a high degree, if localization is provided by a random potential: why would this favor emission from, e.g., R_{hh} high-symmetry points, which would have the right selection rules and a roughly fitting g factor based on theory calculations (Ref. 21)?

Response 3.3) We thank the reviewer for this thoughtful question. To respond we carried out additional calculations on the optical oscillator strength with a spaced TMD heterostructure which indeed explains this effect. To understand the opposite selection rules in the DC and hBN-separated regions, we must consider both the effect of the moiré potential as well as the stacking dependent IX oscillator strength. The oscillator strength was calculated as a function of interlayer stacking and separation (see Extended Data Fig. 3). We find that that the oscillator strength is actually dominated by recombination at the R_h^h site which results in primarily observing co-circularly polarized PL. However, in the DC region, the moiré potential traps IXs at the R_h^x stacking site which has the opposite selection rule, emitting cross-circularly polarized PL. Therefore, it is the trapping of IX at R_h^x sites that results in the cross-circularly polarized PL from the DC contact region, whereas the larger oscillator strength of co-polarized emission results in co-circularly polarized PL in the hBN-separated regions. Note that this does not mean that the IX is localized at the R_h^h site in the hBN-separated region, rather that since this local stacking possesses the highest oscillator strength, it will dominate the PL spectrum.

Comment 3.4) I was also surprised by the suppression of ILE PL in the separated region at slightly higher temperatures. Given the low excitation density used by the authors, I wonder if dipole-dipole interaction scattering the ILE out of the detection spot can really be the driving factor for this observation. This could be verified by performing power-dependent PL measurements on the separated region and imaging the emission spot size as a function of power.

Response 3.4) We thank the reviewer for this suggestion. To respond, we performed power dependent PL imaging on the separated region as shown in Figure R2 (for high power). The circle represents the location of the excitation laser spot in the hBN separated region. The sample regions are outlined. Please also see the attached movie which shows the IX flow as function of power. As the power is increased the IXs flow from the hBN separated region to the DC contact region. Figure R2b shows the integrated counts from each region as a function of power. It is clear that the hBN separated region saturates at low power providing evidence of IX flow out of the hBN separated region. We have also provided a .gif movie to show the power dependence of the imaged PL intensity.

Figure R2: Power dependent flow of IX. **a**, PL image of the sample when the hBN separated region is excited with high power ($112 \mu\text{W}$) 714 nm laser. The red circle shows the excitation spot position. The hBN separated region is outlined in blue and the direct contact area is shown in red. By increasing the power, the IXs flow from the hBN separated region to the DC area. **b**, Integrated PL intensity of Fig. R2a as a function of power shows the IX starts to flow to the DC region from the hBN separated region at low power. By increasing the power to $40 \mu\text{W}$ the intensity from the DC region is equal to hBN separated region. Above $40 \mu\text{W}$, the majority of IXs flow to the DC area while the PL in hBN separated region saturates.

Comment 3.5) At the current stage, I do not feel that the manuscript provides fully convincing

evidence to support the authors claims, but a suitable revision addressing the above-mentioned points could be of great interest to the 2D materials community and warrant publication in Nature communications.

Response 3.5) We thank the reviewer again for their constructive review and hope that they feel that the improved manuscript is now suitable for publication.

1. Seyler, K. L. *et al.* Signatures of moiré-trapped valley excitons in MoSe₂/WSe₂ heterobilayers. *Nature* **567**, 66–70 (2019).
2. Joe, A. Y. *et al.* Electrically controlled emission from singlet and triplet exciton species in atomically thin light-emitting diodes. *Physical Review B* **103**, 161411 (2021).
3. Woźniak, T., Faria Junior, P. E., Seifert, G., Chaves, A. & Kunstmann, J. Exciton g factors of van der Waals heterostructures from first-principles calculations. *Physical Review B* **101**, 235408 (2020).
4. Ciarrocchi, A. *et al.* Polarization switching and electrical control of interlayer excitons in two-dimensional van der Waals heterostructures. *Nature Photonics* **13**, 131–136 (2019).
5. Rosenberger, M. R. *et al.* Twist Angle-Dependent Atomic Reconstruction and Moiré Patterns in Transition Metal Dichalcogenide Heterostructures. *ACS Nano* **14**, 4550–4558 (2020).

Reviewers' Comments:

Reviewer #2:

Remarks to the Author:

The revised version of this manuscript answers all my concerns satisfactorily. So, I recommend the paper be published in Nature Communications.

Reviewer #3:

Remarks to the Author:

The authors have thoroughly addressed my questions in their response and revision of the manuscript. Their calculations of the oscillator strengths, and the very nice power-dependent PL measurements strengthen their arguments.

I fully support publication of the manuscript at Nature Communications in its present form.

Response to Reviewers (NCOMMS-22-18201-T)

Reviewer #2 (Remarks to the Author):

The revised version of this manuscript answers all my concerns satisfactorily. So, I recommend the paper be published in Nature Communications.

Response: We thank the reviewer for their thoughtful review and support.

Reviewer #3 (Remarks to the Author):

The authors have thoroughly addressed my questions in their response and revision of the manuscript. Their calculations of the oscillator strengths, and the very nice power-dependent PL measurements strengthen their arguments.

I fully support publication of the manuscript at Nature Communications in its present form.

Response: We thank the reviewer again for the useful feedback that further increased the quality of the paper.